# JAXA Level 2 cloud and precipitation microphysics retrievals based on EarthCARE radar, lidar and imager: The CPR_CLP, AC_CLP, ACM_CLP products

Kaori Sato[1], Hajime Okamoto[1], Tomoaki Nishizawa[2], Yoshitaka Jin[2], Takashi Y. Nakajima[3], Minrui Wang[3], Masaki Satoh[4], Woosub Roh[4,5], Hiroshi Ishimoto[6], Rei Kudo[6]

[1] Research Institute for Applied Mechanics, Kyushu University, Fukuoka, 816-8580, Japan
[2.]Earth System Division, National Institute for Environmental Studies, Tsukuba, 305-8506, Japan
[3.]Research & Information Center, Tokai University, Kanagawa, 2591292, Japan
[4.]Atmosphere and Ocean Research Institute, The University of Tokyo, Chiba, 2778564, Japan
[5.]Tokyo University of Marine Science and Technology, Tokyo, 1358533, Japan
[6] Meteorological Research Institute, Japan Meteorological Agency, Tsukuba, 305-0052, Japan

*Correspondence to*: Kaori Sato (sato@riam.kyushu-u.ac.jp)

**Abstract**

This study introduces the primary products and features of active sensor-based Level 2 cloud microphysics products of the Japanese Aerospace Exploration Agency (JAXA; i.e., The cloud radar standalone cloud product (CPR_CLP), the radar-lidar synergy cloud product (AC_CLP), and the radar-lidar-imager cloud product (ACM_CLP)). Combined with 94-GHz Doppler cloud profiling radar (CPR), 355-nm high-spectral-resolution lidar (Atmospheric Lidar: ATLID) and Multi-Spectral Imager (MSI), these products provide a detailed view of the transitions of cloud particle categories and their size distributions. Simulated EarthCARE Level 1 data mimicking actual global observations were used to assess the performance of the JAXA Level 2 cloud microphysics product. Evaluation of the product revealed that the retrievals reasonably reproduced the vertical profile of the modelled microphysics. Further validation of the products is planned for post-launch calibration/validation. JAXA Level 2 velocity-related products (i.e., CPR_VVL, AC_VVL, and ACM_VVL) such as hydrometeor fall speed and air vertical velocity will be described in a future paper.

## 1. Introduction

With advances in high-resolution global cloud-resolving models for climate simulations, there is increasing interest in the observation of global air vertical velocity and cloud property information. Air vertical velocity distributions are important for hydrometeor formation (Sullivan et al., 2016) and cloud dynamcis, and EarthCARE will provide the first dense global observations. A method for the simultaneous retrieval of air vertical velocity, particle sedimentation velocity, and microphysics using similar variables obtainable by EarthCARE cloud profiling radar (CPR) and atmospheric lidar (ATLID) has been developed and tested using the Equatorial Atmospheric Radar (Sato et al., 2009). Information derived using this

method was used to investigate ice water content in relation to convective activity to evaluate an atmospheric general circulation model (Sato et al., 2010). It is anticipated that analyses of EarthCARE data will be useful for quantifying the role of air vertical velocity in determining cloud properties and lifetime. The lidar depolarization measurement is a strong indicator of particle phase, shape, and orientation (Yoshida et al., 2010). Radar-lidar synergy algorithm with a specular reflection mode investigated the mass mixing ratio of oriented plates (2D types) and randomly oriented crystals (3D ice) within clouds (Okamoto et al., 2010) and ice precipitation (Sato and Okamoto, 2011) at each vertical grids from CloudSat and Cloud Aerosol Lidar and Infrared Pathfinder Satellite Observations (CALIPSO) data. The recent development of numerical simulations of lidar backscattering for interpreting 355-nm high-spectral resolution polarization lidar (HSRL) measurements has demonstrated the possibility of deriving more specific ice habit category information from EarthCARE, in addition to the cloud phase and 2D/3D ice category. Measurements from ground-based HSRL support such theoretical studies (Jin et al., 2020, 2022). These unique aspects are incorporated into active sensor-based Japanese Aerospace Exploration Agency (JAXA) Level 2 (L2) cloud algorithms to create products that beneficial for investigating cloud formation and cloud-precipitation processes. A preliminary study of the JAXA L2 cloud product using available satellite data produced exciting results, displaying a unique geographical preference for the occurrence and height-dependent characteristics of different ice habit categories (Sato and Okamoto, 2023). Each component of the EarthCARE JAXA L2 products should significantly increase our understanding of the coupling of cloud microphysics, radiation, and dynamics.

This paper is organized as follows. Section 2 provides an overview of active sensor-based JAXA L2 cloud products and simulated EarthCARE Level 1 (L1) data. The JAXA L2 cloud product is demonstrated and assessed in Section 3. Section 4 summarizes the results and outlines future expectations for EarthCARE that has been successfully launched into orbit on 28 May (15:20 local time).

## 2. Data and description

### 2.1 Overview of JAXA Level 2 Cloud Microphysics Products

#### 2.1.1 Primary cloud products

Standard cloud property (CLP) products (i.e., CPR standalone CPR_CLP product, CPR-ATLID synergy AC_CLP product, and CPR-ATLID-MSI synergy ACM_CLP product) include a cloud mask, cloud particle type, cloud particle habit category, cloud microphysics, cloud optical thickness, and cloud water/ice paths (Table 1). The microphysical properties of all hydrometeor types in the standard products are reported in the cloud microphysics product, and precipitation-sized particles are not separated into precipitation products. JAXA L2 research cloud products include velocity-related products such as sedimentation velocity and air vertical velocity (Sato et al., 2009), which are designated CPR_VVL, AC_VVL, and ACM_VVL; precipitation-only products (e.g., rain and snow rates; CPR_RAS, AC_RAS, and ACM_RAS) (Table 1). Details of these research products will be reported in a future paper. All products are reported using the Joint Standard Grid

(JSG) with 1-km horizontal and 100-m vertical grid spacing. Note that CPR_CLP, AC_CLP, and ACM_CLP are produced with and without the use of L2 CPR Doppler velocity to show the effect of additional information obtained from Doppler velocity. The version without Doppler velocity will eventually be updated based on the version using Doppler velocity. Similarly, research products will be developed through RAS and VVL, and results fulfilling the release criteria may be added to the standard products (i.e., CPR_CLP, AC_CLP, and ACM_CLP) for release.

### 2.1.2 Rationale for producing three products

The CPR standalone (CPR) algorithm is considered to produce the simplest and most stable products, which are not affected by the observation and retrieval performance of other sensors, but with relatively higher uncertainty due to the small number of observables. The CPR–ATLID synergy (AC) cloud algorithm, and the CPR–ATLID–MSI (ACM) algorithm are generally considered to produce more reliable estimates of cloud microphysics and can handle more complicated scenes in terms of cloud phase with more observables and greater sensitivity. Notably, the degree of improvement in multi-sensor retrievals can be affected by many factors (e.g., day/night differences in ATLID and MSI observations).

The JAXA L2 cloud microphysics algorithms for the CPR standalone, 2-sensor, and 3-sensor synergy products share the same basic algorithms and assumptions. Less synergetic algorithms are developed and trained with more synergetic algorithms (e.g., the CPR standalone algorithm relative to 2- and 3-sensor algorithms, and the 2-sensor algorithm relative to the 3-sensor algorithm). A comparison of the three products and careful investigation of the causes underlying differences in the retrieval results according to different synergy levels will contribute to the development of better algorithms and more reliable global cloud microphysical products. The release of these three products by JAXA supports the development of retrieval algorithms allowing for the consistent treatment and integration of comprehensive long-term, spatially dense observations from active sensors on various platforms with differing sensitivity levels to create homogenous microphysics data. Collocated lidar and cloud radar measurements will not always be possible in future missions; therefore, single-sensor algorithms that are consistent with synergetic algorithms are needed (e.g., to process cloud radar data from CloudSat, EarthCARE, and future missions with single CPR measurements).

### 2.1.3. Summary of available information, challenges, general approaches, and additional information used to constrain retrievals

For cloud microphysics, CPR_CLP and ACM_CLP share the same basic algorithm architecture as AC_CLP, whereas in CPR_CLP, the ATLID observables are simulated based on observations to drive AC_CLP-like retrieval. ACM_CLP has additional steps to handle inputs from the MSI. Further, the framework of ice and water microphysics retrieval algorithms have similar structure. For these algorithms, a maximum of two size modes in each JSG are used to treat coexistence of cloud ice and snow in the ice phase, cloud liquid and ice (or snow) in the mixed phase, and cloud liquid and liquid precipitation in the liquid phase. Cloud ice microphysics are generally retrieved by CPR-ATLID synergy, whereas ice and liquid precipitation are often retrieved by CPR alone due to the attenuation of ATLID signals, and cloud liquid is

retrieved through either ATLID-only or CPR-only retrieval schemes, as lidar and cloud radar are considered to be sensitive to different portions of the particle size distribution, particularly for water clouds.

Cloud microphysics retrieval in CPR-only regions involves challenges in producing effective radius ($r_{eff}$) and ice water content (IWC) or liquid water content (LWC) solely from radar reflectivity ($Z_e$) constrained by pulse-integrated attenuation (PIA) when Doppler velocity is not used. The dependence of $Z_e$ on cloud microphysical properties reflects cloud physical processes (e.g., Khain et al., 2008). A single size mode cannot explain the transition stage between cloud and precipitation (Krasnov and Russchenberg, 2002). Therefore, a methodology to consider two size modes in each JSG is developed for a better interpretation of $Z_e$ profiles in both ice- and liquid-clouds. $Z_e$ is less sensitive to cloud particles in the presence of precipitation particles in ice- or liquid clouds, and $Z_e$ is less sensitive to liquid cloud particles in the presence of ice particles in mixed phased clouds. In such cases, the additional information of MSI optical thickness is effective for constraining cloud $r_{eff}$ and LWC (or IWC) derived from AC_CLP in the ACM_CLP scheme. For CPR_CLP, the same microphysics retrieval scheme employed by AC_CLP for the CPR-only detected cloud region is used. To run the AC_CLP scheme, the statistical relationships between lidar observables and $Z_e$ for the water and ice phases are derived from CALIPSO and CloudSat long-term observations and applied to create ATLID-like observations (Okamoto et al., 2020) as a function of $Z_e$ that is fully attenuated in optically thick regions, realistically recreating observations. The current version of ATLID-like inputs will be replaced by inputs directly derived from ATLID and CPR observations. Currently, the ATLID-like input is used for only for the ice phase. For liquid cloud microphysics, ATLID-only and CPR-only retrievals are obtained and combined in the AC_CLP algorithm due to the differing sensitivity of the sensors to cloud particle size. For CPR_CLP, the CPR-only retrieval without the ATLID-like input is conducted for liquid cloud microphysics.

| Standard Product | Description | L2a CPR CPR_CLP* | L2b CPR-ATLID AC_CLP* | L2b CPR-ATLID-MSI ACM_CLP** |
|---|---|---|---|---|
| cloud mask | cloud and precipitation (described in Okamoto et al., 2024a) | ✔ST | ✔ST | ✔ST |
| cloud particle type | clear/warm water/ supercooled water /3d ice /2d plate /mixture of 3d ice and 2d plate / liquid drizzle / mixed-phase drizzle / rain / snow /water+liquid drizzle / water+rain / mixed-phase / melting layer | ✔ST | ✔ST | ✔ST |
| cloud particle category | cloud particle habit categories (2Dplate, 2Dcolumn, bullet rosette/3Daggregates, droxtal/compact, voronoi/irregular, fractal, liquid-phase types from cloud particle type product) | ✔ST | ✔ST | ✔ST |
| cloud water effective radius | cloud and precipitation both liquid-phase and ice-phase microphysics are reported at each vertical grid (precipitation-only products are provided by CPR_RAS, AC_RAS, ACM_RAS) | ✔ST | ✔ST | ✔ST |
| cloud ice effective radius | | ✔ST | ✔ST | ✔ST |
| cloud water content | | ✔ST | ✔ST | ✔ST |
| cloud ice content | | ✔ST | ✔ST | ✔ST |
| total cloud water number concentration | | ✔ST | ✔ST | ✔ST |
| total cloud ice number concentration | | ✔ST | ✔ST | ✔ST |
| - cloud effective radius 1 - cloud effective radius 2 | subcategory products to infer particle size distribution of cloud and precipitation | ✔ST | ✔ST | ✔ST |
| - cloud water content 1 - cloud water content 2 | cloud phase 1 and 2 : (0) not retrieved, (1) water, (2) ice, (-9) clear combination of ice+ice, water+water, ice+water are possible | ✔ST | ✔ST | ✔ST |
| - cloud number concentration 1 - cloud number concentration 2 | cloud effective radius, water content, and number concentration corresponding to cloud phase 1 and 2 are reported at each vertical grid | ✔ST | ✔ST | ✔ST |
| - cloud phase1 - cloud phase2 | | ✔ST | ✔ST | ✔ST |
| optical thickness | liquid + ice phase | ✔ST | ✔ST | ✔ST |
| cloud water path | | ✔ST | ✔ST | ✔ST |
| cloud ice water path | | ✔ST | ✔ST | ✔ST |
| **Research Product** | **Description** | **CPR_RAS** | **AC_RAS** | **ACM_RAS** |
| rain and snow properties (rain/snow rate, rain/snow water | vertical profile | ✔ER | ✔ER | ✔LR |
| | | | **AC_MRA** | |
| mass ratios of 2D plates | Okamoto et al., (2010) | | ✔ER | |
| | | **CPR_VVL** | **AC_VVL** | **ACM_VVL** |
| cloud doppler velocity | In-cloud Doppler velocity with folding correction after applying cloud mask | ✔ER | ✔ER | ✔LR |
| total cloud terminal velocity | Mean terminal velocity of cloud and precipitation particles | ✔ER | ✔ER | ✔LR |
| - cloud terminal velocity 1 - cloud terminal velocity 2 | Terminal velocity corresponding to (1) and (2) | ✔ER | ✔ER | ✔LR |
| air vertical velocity | In-cloud air vertical velocity | ✔ER | ✔ER | ✔LR |

\* CPR_CLP and AC_CLP standard products will be updated with the use of Doppler Velocity
\*\* CPR-ATLID-MSI standard cloud property products with the use of Doppler velocity are provided by ACM_CDP as research products (LR)

**Table 1: Primary parameters of the Japanese Aerospace Exploration Agency (JAXA) Level 2 (L2) standard (ST) and research (ER/LR) active sensor-based cloud products, including the standalone products (CPR_CLP, CPR_RAS, and CPR_VVL), CPR–ATLID synergy cloud products (AC_CLP, AC_RAS, and AC_VVL), and CPR–ATLID–MSI products (ACM_CLP, ACM_RAS, and ACM_VVL). The ER and LR products are processed by the JAXA Earth Observation Research Center Research and Application System and JAXA Laboratories, respectively. CPR_CLP and AC_CLP standard products will be updated with the use of Doppler velocity, and ACM_CLP cloud property products updated with the use of Doppler velocity will be provided by ACM_CDP as research products (LR). ATLID, atmospheric lidar; CPR, cloud profiling radar; MSI, Multi-Spectral Imager.**

## 2.2 Processing flow of the JAXA Level 2 cloud microphysics product

Figure 1 shows the flow of the L2 cloud products. The JAXA L2 Echo algorithm processes CPR L1 data and was developed by the National Institute of Information and Communications Technology (NICT). The major outputs from the JAXA L2 Echo product to CPR_CLP, AC_CLP, and ACM_CLP are radar reflectivity factor ($Z_e$), Doppler velocity ($V_D$), normalized radar cross-section ($\sigma_0$), pulse integrated attenuation (PIA), gaseous attenuation, clutter mask, and quality flags. The inputs from the JAXA L2 ATLID product (Nishizawa et al., 2024) to the AC_CLP and ACM_CLP algorithms are the L2 ATLID observables (i.e., extinction coefficient $\alpha_{ext}$, attenuated $\beta_{att}$ and true backscattering coefficient $\beta$, and depolarization ratio $\delta$) and their aerosol and cloud components (Kudo et al., 2016; Kudo et al., 2023), ATLID-only cloud mask and cloud type (Okamoto et al., 2024a). Aerosol extinction is used to handle attenuation due to aerosols above the cloud layers.

The L2 cloud algorithms are processed in the following order: CPR_CLP, AC_CLP, and ACM_CLP. The cloud mask, cloud type, and cloud particle category products from each algorithm are passed to the high-order synergy algorithms. The CPR-only cloud mask, cloud type, and cloud particle category products from L2a CPR_CLP are input to the L2b AC_CLP algorithm, and these CPR-only derived products are combined with the ATLID-only cloud mask, cloud type, and cloud particle category to produce synergy CPR-ATLID products. These products are then applied to the AC_CLP algorithm to derive cloud microphysics products. The AC_CLP cloud mask, cloud type, and cloud particle category products are further passed to the ACM_CLP algorithm and used for 3-sensor microphysics retrieval. The MSI is not currently used to improve the cloud mask, type, and category products; therefore, these products from ACM_CLP are the same as those from AC_CLP. The inputs from JAXA L2 MSI products to ACM_CLP are the optical thickness of the ice and liquid phases (Nakajima et al., 2019; Wang et al., 2023), which are used to constrain CPR_CLP and AC_CLP microphysics estimates. The JAXA Level 2 cloud product is further handled by the JAXA L2 four-sensor radiation products (Yamauchi et al., 2024). Details of the relationships among JAXA Level 2 algorithms and products were provided by Okamoto et al. (2024b).

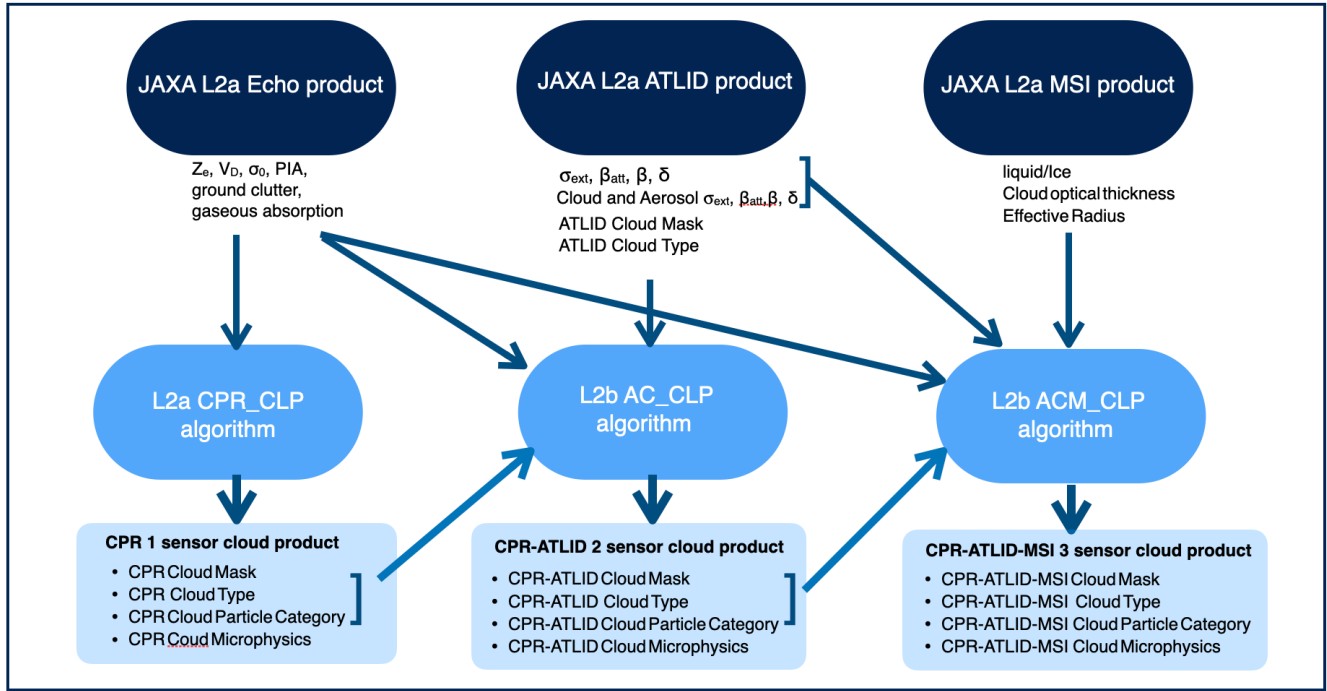

Figure 1: Flow of the Japanese Aerospace Exploration Agency (JAXA) Level 2 (L2) cloud products, including the standalone product (CPR_CLP), the CPR–ATLID synergy cloud product (AC_CLP), and the CPR–ATLID–MSI product (ACM_CLP). JAXA L2 Echo product contains the radar reflectivity factor ($Z_e$), Doppler velocity ($V_D$), normalized radar cross-section ($\sigma_0$), pulse integrated attenuation (PIA). JAXA L2 ATLID product contains the extinction coefficient ($\alpha_{ext}$), attenuated ($\beta_{att}$) and true backscattering coefficient ($\beta$), and depolarization ratio ($\delta$). ATLID, atmospheric lidar; CPR, cloud profiling radar; MSI, Multi-Spectral Imager.

## 2.3 Description of the JAXA Level 2 cloud microphysics product

The following section provides a brief overview and highlights of the standard JAXA L2 cloud microphysics products.

### 2.3.1 Preprocessing for cloud microphysics retrieval

### 2.3.1.1 Cloud mask

The ATLID-only cloud mask is processed by the ATLID_CLA algorithm (Nishizawa et al., 2024), the CPR-only cloud mask is processed by the CPR_CLP algorithm (Okamoto et al., 2024a), and the MSI-only cloud mask is processed by the MSI_CLP algorithm (Nakajima et al., 2019). For ATLID, aerosol, cloud, and surface components are discriminated from clear pixel when the Mie backscattering coefficient is significant compared to the noise level (Nishizawa et al., 2024). A

cloud mask scheme is then applied; this scheme includes a vertically variable threshold value for the Mie backscattering coefficient (or particle backscattering coefficient when the Rayleigh backscattering coefficient is significant), as well as a spatial continuity test to exclude noisy pixels. The lack of sufficient surface signal is used to identify fully attenuated ATLID pixels below aerosol or cloud layers. Similarly, the CPR cloud mask scheme considers noise level, continuity testing, and surface echo information to determine sufficient radar echo power for cloud and precipitation analysis, as well as full attenuation of the radar signal. The AC_CLP synergy cloud mask scheme merges the single active sensor cloud mask results from ATLID_CLA and CPR_CLP, and then flags cloudy pixels in ATLID, CPR, or both. MSI cloud mask information is not used for the ACM_CLP cloud mask. The AC_CLP and ACM_CLP cloud mask products are currently identical.

### 2.3.1.2 Cloud type

The ATLID cloud type scheme (ATLID_CLA) uses $\delta$, $\beta_{att}$, and temperature to identify the cloud phase and ice particle orientation, which is designated as two-dimensional (2D) ice, three-dimensional (3D) ice or mixed 2D and 3D ice (Okamoto et al., 2024a). The CPR cloud type scheme (CPR_CLP) uses mainly $Z_e$ (along with its vertical profile) and temperature to discriminate the hydrometeor phase, ice particle orientation, precipitation type (snow, drizzle, or rain) and melting layer (Okamoto et al., 2024a). The AC_CLP synergy cloud type scheme combines ATLID_CLA with CPR_CLP and reclassifies the cloud type when estimates from the two sensors differ according to the classification rule specified by Kikuchi et al. (2017). The differing particle size sensitivity of CPR and ATLID aid in the identification of mixed-phase clouds and mixed cloud-precipitation types (i.e., cloud water + drizzle or cloud water + rain). The ACM_CLP and AC_CLP cloud types are identical. In addition, Doppler velocity will be used to improve differentiation between snow and rain and between cloud and drizzle. Further details of the cloud mask and cloud particle type products were reviewed by Nishizawa et al. (2024) and Okamoto et al. (2024a).

### 2.3.1.3 Cloud particle category (CPC)

After applying the cloud mask and cloud phase discrimination schemes (Okamoto et al., 2024a), one of the main products of the EarthCARE JAXA L2 cloud product is the cloud particle category product, which enables more detailed comprehensive exploration of the ice particle habit category contained within each JSG grid. Among cloud particle categories, the liquid-phase types are the same as those in the cloud type product (subsection 2.3.1.2). Ice particles are further categorized based on ATLID lidar ratio and depolarization ratio diagrams (Okamoto et al., 2019; 2020; Sato and Okamoto, 2023). This information is anticipated to be instrumental for general remote sensing applications (Van Diedenhoven, 2018; Letu et al., 2016) and the development of ice optical parameterization (Li et al., 2022) and hydrometeor sedimentation velocity parameterization for use in numerical models. The retrieved ice particle habit categories include horizontally oriented 2D plates and their assemblages, 2D columns and their assemblages, bullet rosettes and 3D-oriented aggregate types, droxtal/compact types, Voronoi/irregular/roughened types, and fractal-type snow aggregates (Ishimoto et al., 2008, 2012). ATLID-only CPC is used to train the CPR-based algorithm for ice particle category retrieval from $Z_e$ and

temperature information in regions with CPR-only measurements. The CPR-only CPC product is obtained from CPR_CLP. CPR_CLP, and ATLID-only CPC are combined to produce the synergy AC_CLP CPC product. For ice categories, ATLID-only CPC estimates are used when both CPR_CLP CPC and ATLID-only CPC estimates are available for the same JSG grid. The Doppler velocity will be further used to improve category identification, particularly for snow types (e.g., graupel or hail).

### 2.3.2 Cloud microphysics

In CPR_CLP, ACP_CLP, and ACM_CLP, forward models corresponding to the derived cloud particle categories are used to analyze the observations from each sensor, and microphysics corresponding to each category are thus obtained. The single scattering properties of ice particles with various shapes and orientations are calculated using physical optics (Borovoi et al., 2012) and modified geometrical optics integral equation methods (Masuda et al., 2012) for ATLID specification (Okamoto et al., 2019), and discrete dipole approximation and finite-difference time domain (FDTD) methods for CPR wavelength (Sato et al., 2011; Ishimoto et al., 2008, 2012); Mie theory is used for the liquid phase and multiple scattering effects are estimated based on Sato et al. (2018, 2019).

The total effective radius for cloud and precipitation information is given as:

$$r_{\text{eff}} = \int r_{eq}^3 \frac{dn(r_{eq})}{dr_{eq}} dr_{eq} \bigg/ \int r_{eq}^2 \frac{dn(r_{eq})}{dr_{eq}} dr_{eq} \qquad (1)$$

where $r_{eq}$ is the melted mass equivalent radius to a sphere, $dn/dr_{eq}$ is the size distribution function. For both ice- and liquid-phase clouds, a maximum of two different particle size distributions (i=1,2) can be considered within one JSG grid to handle the presence of cloud and precipitation modes, i.e., $\frac{dn(r_{eq})}{dr_{eq}} = \sum_{i=1}^{2} \frac{dn_i(r_{eq})}{dr_{eq}}$. The corresponding effective radius is given as:

$$r_{\text{eff,i}} = \int r_{eq}^3 \frac{dn_i(r_{eq})}{dr_{eq}} dr_{eq} \bigg/ \int r_{eq}^2 \frac{dn_i(r_{eq})}{dr_{eq}} dr_{eq} \quad (i = 1,2) \qquad (2)$$

For $dn_i/dr_{eq}$, a modified gamma size distribution,

$$\frac{dn_i(r_{eq})}{dr_{eq}} = \frac{N_{o,i}}{\Gamma(p)r_{m,i}} \left(\frac{r_{eq}}{r_{m,i}}\right)^{p-1} exp\left(-\frac{r_{eq}}{r_{m,i}}\right) \quad (i = 1,2) \qquad (3)$$

in which $r_m$ is the characteristic radius and the dispersion value is p = 2 (Okamoto, 2002; Sato and Okamoto, 2011), is employed for cloud ice, snow, and rain in cold precipitation. A log-normal size distribution,

$$\frac{dn_i(r_{eq})}{dr_{eq}} = \frac{N_{o,i}}{\sqrt{2\pi}r_{eq}ln\sigma} exp\left\{-\frac{[ln(r_{eq}/r_{o,i})]^2}{2(ln\sigma)^2}\right\} \quad (i = 1,2) \qquad (4)$$

in which $r_o$ is the mode radius and the standard deviation of the distribution is σ = 1.5 (Okamoto, 2002), is assumed for warm water, super-cooled liquid, and warm precipitation.

In the following, general approaches for cloud microphysics retrievals are explained based on the AC_CLP cloud microphysics algorithm, which are common to CPR_CLP and ACM_CLP cloud microphysics algorithms.

### 2.3.2.1 Ice cloud microphysics

For ice clouds, a lidar-only cloud region, lidar–radar overlap cloud region, and radar-only region generally exist for ice and liquid precipitation. An algorithm to retrieve microphysical properties that considers a mixture of two particle types at maximum (i.e., 2D and 3D ice) has been developed for ice cloud regions observed with CloudSat and CALIPSO synergy (Okamoto et al., 2010) using $Z_e$, the attenuated backscattering coefficient $\beta$, and the depolarization ratio. A framework to extend the applicability of the microphysics retrieval algorithm from the cloud region to the entire precipitation region in the vertical column was developed to efficiently reflect information from the lidar–radar overlap region to the microphysics retrieved in the CloudSat- or CALIPSO-only region (Sato et al., 2011, 2020). The relationships between microphysical properties ($r_{eff}$ and IWC) and $\beta$ or $Z_e$ in the vertical cloud grids of the lidar–radar overlap region were derived for each profile and used to estimate the microphysical properties in the radar- or lidar-only cloud region (Sato et al., 2011). The EarthCARE JAXA L2 cloud microphysics retrieval algorithms extend these algorithms in the following three aspects: (1) the spatial variability of the microphysics and observables are considered to derive more reliable relationships among cloud microphysics and observables, (2) the microphysics estimates in the ice precipitation region far from the lidar–radar overlap region of a precipitation system are further improved by extending the microphysics estimates from the precipitation region upward rather than downward from the lidar–radar region (Heymsfield et al., 2018), and (3) single-size mode for cloud ice is considered for lidar-only cloud region and lidar–radar overlap cloud region, while two different size modes for cloud ice and ice precipitation (snow) are considered for the CPR-only region existing from the bottom altitude of the lidar–radar overlap region to the top altitude of the melting level. The PIA is used to correct the attenuation of $Z_e$. (Iguchi et al., 2000).

Specifically, for (1), the L2 cloud microphysics algorithm uses $r_{eff}$ and IWC for all horizontal and vertical grids within the radar–lidar overlap region embedded in each cloud system to obtain robust relationships of cloud microphysics with $Z_e$ and $\beta$ (e.g., $Z_e$–IWC relationships, $Z_{e,1} = a_1 IWC_1^{b1}$ are determined for each record, where $Z_e$ [mm$^6$ m$^{-3}$] and IWC [g m$^{-3}$]). These relationships are derived for each record using all data within each cloud system (or within a single EarthCARE orbit frame when a sufficient number of points cannot be obtained to derive the statistics) weighted by distance from the target profile record and are used to provide initial estimates of cloud ice microphysics based on $Z_e$ or $\beta$ in the CPR-only (ice cloud and ice precipitation) or ATLID-only (ice cloud) regions, respectively.

For (2), the relationship between the microphysics and observables is expected to change from the cloud region to the precipitation region. Because lidar signals are fully attenuated at optically thick precipitation region, new relationships for ice precipitation are derived using CPR data. In this process, CPR data at melting levels or layers around the ice–liquid interfaces of a precipitation system are used. At the top of the melting level, it is assumed that only precipitation mode exists ($Z_e = Z_{e,2}$), and during melting, the mass in each size bin (i.e., $r_{eff}$) remains constant across several successive layers (Heymsfield et al., 2018). For a given $r_{eff}$, dBZ$_e$ changes due to the different scattering properties for ice and liquid. Therefore, $r_{eff}$ and IWC (or LWC) are derived and the relationships ($Z_{e,2} = a_2 IWC_2^{b2}$) can be established for ice precipitation (snow) holding the coefficient $b_2$ at the value derived in (1) ($b_2 = b_1$) for each record.

For (3), $Z_{e,1}$ and $Z_{e,2}$ for the two size modes (cloud ice and snow) in the CPR-only ice precipitation region at each vertical grid ($Z_{e,1} + Z_{e,2} = Z_e$) are determined as follows. The ratio $IWC_2/(IWC_1+IWC_2) = IWC_2/IWC = A$ increases linearly from 0 at the bottom of the lidar–radar overlap region to 1 at the top of the melting level. $A$ is given as, $A = \int_h^{ht} Z_e \, dh / \int_{hm}^{ht} Z_e \, dh$, with a range of 0 to 1, where the integrated $Z_e$ from the bottom altitude of the lidar–radar overlap region ($h_t$) to a certain altitude h below $h_t$ within the CPR-only ice precipitation region ($\int_h^{ht} Z_e \, dh$) is normalized using the value integrated to the melting level altitude $h_m$ ($\int_{hm}^{ht} Z_e \, dh$). As the $Z_e$–IWC relationships for both cloud ice and snow are derived, determining the vertical profile of $IWC_2/IWC$ is equivalent to providing the relationship between $Z_{e,1}$ and $Z_{e,2}$ for each vertical grid. Therefore $IWC_i$, $r_{eff,i}$ (i=1,2) and other microphysical properties are derived for each JSG grid (Table 1).

In microphysics retrieval for convective/stratiform rain below the melting level, only the precipitation size mode is assumed to exist. The $r_{eff}$ and LWC obtained at the rain top altitude of each observation record described in (2) are used to derive the No and x values of the Marshall–Palmer size distribution ($dn/dD = No \, e^{-\Lambda D}$ [m$^{-4}$], where D is the particle diameter, $\Lambda = xR^{0.21}$ [cm$^{-1}$] and R is the rain rate in mm/hr, which is a function of LWC and $r_{eff}$) (Marshall and Palmer, 1948). No and x are assumed to be constant within the vertical profile for rain in a given record and are used to determine the vertical profiles of LWC and $r_{eff}$ for the modified gamma size distribution associated with each $Z_e$ value in the rain region.

Generally, for the same $Z_e$, when the mass mixing ratio of the small mode to total IWC is overestimated (underestimated), optical thickness will be overestimated (underestimated); in the 3-sensor ACM_CLP algorithm, the mass mixing ratio of the two size modes is further constrained by the optical thickness obtained from the MSI. When only a single size mode is present, the $r_{eff}$ and IWC of the single mode are adjusted to be consistent with MSI optical thickness retrievals. Doppler velocity is expected to effectively improve particle sizing in regions of ice and liquid precipitation, as well as in the breakup of large snow particles during melting (e.g., Fujiyoshi et al., 2023).

**2.3.2.2 Liquid cloud microphysics**

A two-size-mode approach similar to the ice cloud microphysics retrieval process is used for water clouds, which considers the coexistence of cloud particles and drizzle. CPR_CLP derives the liquid microphysics corresponding to each size mode from CPR-only scheme. In AC_CLP and ACM_CLP, for JSG grids with ATLID observables, ATLID $\delta$ and $\beta_{att}$ (or $\sigma_{ext}$) are used to derive $r_{eff,1}$ and $LWC_1$ for cloud water or super-cooled water (Sato et al., 2018, 2019; Sato and Okamoto, 2020). As ATLID is expected to provide a better estimate of the cloud mode than CPR, for the CPR and ATLID overlap region, the ATLID cloud microphysics and $Z_{e,1}$ estimate are used for microphysics estimation of the drizzle mode.

In water clouds, *in situ* and ground-based radar measurements have shown that cloud particles and drizzle-sized particles can coexist above −35 dB$Z_e$ (Baedi et al., 2000). Except at very small (< −35 dB$Z_e$) and large values of $Z_e$, where only a single mode is likely to occur, the cloud mode can dominate LWC and $r_{eff}$, whereas the precipitation mode can dominate $Z_e$ (Baedi et al., 2000; Krasnov and Russchenberg, 2005). For this reason, in general, the dependence of total LWC

on $Z_e$ differs significantly from results derived for only cloud particles ($LWC_1$ and $Z_{e,1}$) or only drizzle-sized particles ($LWC_2$ and $Z_{e,2}$) (Baedi et al., 2000). PIA is sensitive to total LWC, and in the CPR-only microphysics retrieval scheme, the $Z_e$–LWC relationship ($Z_e = a LWC^b$, where $Z_e$ [mm$^6$ m$^{-3}$] and LWC [g m$^{-3}$]) and LWC for the cloud+drizzle mode for the JSG grids within each record are determined from PIA and $Z_e$ assuming that b = 5.17 (Baedi et al., 2000). The power $b_i$ of the $Z_e$–LWC relationship for clouds and drizzles are reported to have similar values and assumed to be fixed (i.e., $b_1 \sim 1.17$; Baedi et al., 2000, Fox and Illingworth, 1997, $b_2 \sim 1.58$; Krasnov and Russchenberg, 2002), while the coefficients $a_i$ in the $Z_e$–LWC relationship could differ between clouds and drizzles by several orders of magnitude reflecting the size distribution difference (Khain et al., 2008). As CPR $Z_e$ is more sensitive to the drizzle mode (i.e., $Z_{e,2}$), the $a_1$ coefficient for cloud mode is assumed to be initially fixed at reported value ($a_1 = 0.015$; Baedi et al., 2000), and $a_2$ is derived for each $Z_e$ and LWC profile, given that $LWC_1 + LWC_2 = LWC$ and $Z_{e,1} + Z_{e,2} = Z_e$. Finally, $Z_{e,i}$, $r_{eff,i}$, $LWC_i$ (i=1,2), and other microphysical properties such as the number concentration and particle fall speed are derived for the two size modes.

The liquid cloud microphysics are further constrained by the ATLID observables for the AC_CLP and ACM_CLP algorithms, and the MSI for the ACM_CLP algorithm. Doppler information will be used to improve the microphysics estimates of the precipitation (drizzle) mode.

### 2.3.3 Intended use of Doppler measurements for air vertical velocity and terminal velocity products

The Doppler velocity is intended to be used in at least two approaches; air vertical velocity will be determined by subtracting the $Z_e$-weighted particle fall speed corresponding to each cloud particle category obtained without the use of Doppler velocity, and simultaneous retrieval of air vertical velocity and microphysics through an approach similar to that described by Sato et al. (2009), which considers the difference between the vertical structures of $Z_e$ (reflecting cloud microphysics) and $V_D$ (which is affected by air vertical velocity and cloud microphysics) to extract the air vertical velocity component.

### 2.4 JAXA joint simulator-derived EarthCARE L1 data

The performance of the JAXA L2 cloud algorithms was tested using simulated EarthCARE L1 orbit data created by the JAXA joint simulator (Roh et al., 2023 and references therein). These L1 data are created using cloud and precipitation scenes generated by the Nonhydrostatic Icosahedral Atmospheric Model (NICAM) (Satoh et al., 2014) at 3.5-km horizontal resolution, and profiles of aerosol species simulated by the NICAM Spectral Radiation–Transport Model. Random errors and noise are added to create CPR and ATLID signals, and the spectral misalignment effect of the visible and near-infrared channels is introduced for the MSI (Roh et al., 2023) to mimic actual observations. The simulated L1 data for an EarthCARE orbit are divided into eight frames, and 15 frames, corresponding to nearly 2 orbits, are simulated to include representative cloud and aerosol scenes around the world. All 15 frames are used to evaluate the JAXA L2 cloud product.

## 3. Demonstration and assessment of JAXA Level 2 cloud product

Figure 2 shows the ice particle category product, which was derived using complementary observations from
Cloud-Aerosol Lidar and Infrared Pathfinder Satellite Observations (CALIPSO) and Cloud-Aerosol Lidar with Orthogonal
Polarization (CALIOP) (Sato and Okamoto, 2023). The CALIPSO data was combined onto the CloudSat grid with a
resolution of 240 m vertically and 1 km horizontally (Kyushu University (KU) CloudSat-CALIPSO Merged Data set data;
Hagihara et al., 2010). Lidar ratio and depolarization ratio information from ATLID may offer a more robust classification of
ice particle categories and orientations, and long-term analysis from CALIOP to ATLID will increase the reliability of the
product (Okamoto et al., 2020). The EarthCARE L2 data will be provided at 100 m vertical resolution.

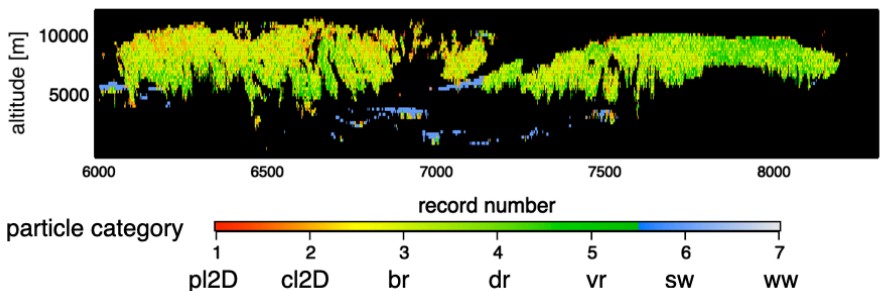

**Figure 2: Demonstration of the JAXA L2 ice particle category product reported from the JAXA L2 ATLID product (ATL–CLA). The dominant ice category type is classified into [1]2D plate (pl2D), [2]2D column (cl2D), [3]3D bullet (bullet rosettes, 3D**
**aggregate category) (br), [4] Droxtals (dr), and [5] Voronoi types (vr), [6] supercooled water (sw), [7] warm water (ww). The JAXA EarthCARE L2 cloud algorithms were modified to be applied to A-Train data, and the ice category classification was derived using Cloud-Aerosol Lidar with Orthogonal Polarization (CALIOP).**

Cloud microphysics retrieved from the simulated EarthCARE L1 data are compared with the truth. For the
comparison, we used the AC_CLP standard products. Figure 3 shows an example of the time–height cross-section of the
simulated CPR measurements and the ATLID L2a cloud backscatter for a cirrus case (scene 1), snow precipitation (scene 2),
and a liquid-phase cloud scene (scene 3). Overall, there was good consistency between the simulated and retrieved cloud
water contents and effective radius, where the AC_CLP standard retrievals reasonably reproduced vertical variation in the
microphysical properties seen in the model (Fig. 4-6).

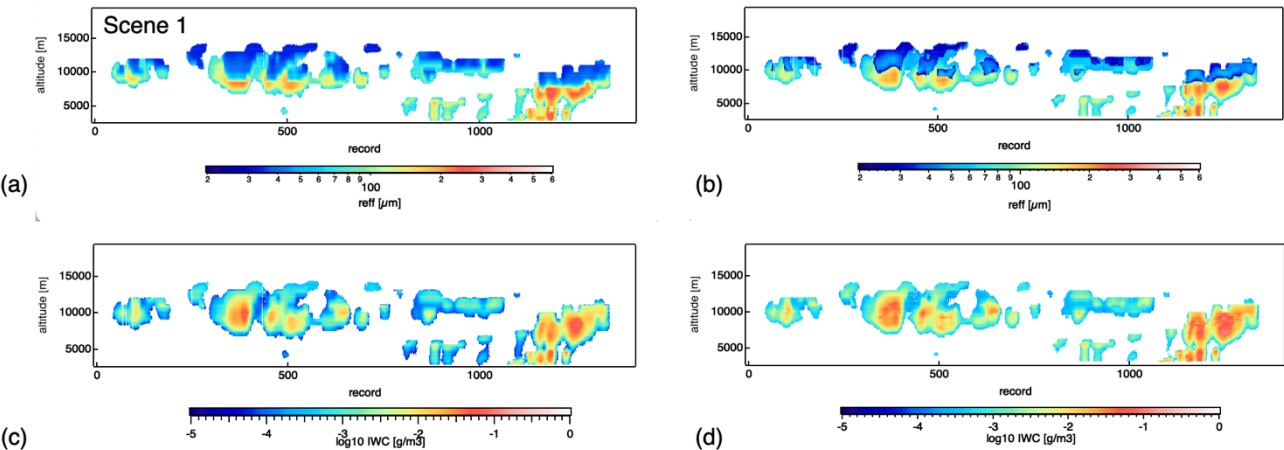

**Figure 3: Inputs of CPR Ze measurements (left column) simulated using the JAXA joint simulator and ATLID L2a cloud backscatter product (right column) for different cloud scenes. (a)(d) Scene 1 is a cirrus case, (b)(e) scene 2 is a case with more ice precipitations, and (c)(f) scene 3 is dominated by liquid-phase.**

**Figure 4: Time–height cross-section of (a, c) simulated and (b, d) retrieved effective radius and total water content for ice-phase corresponding to scene 1 in Fig.4(a).**

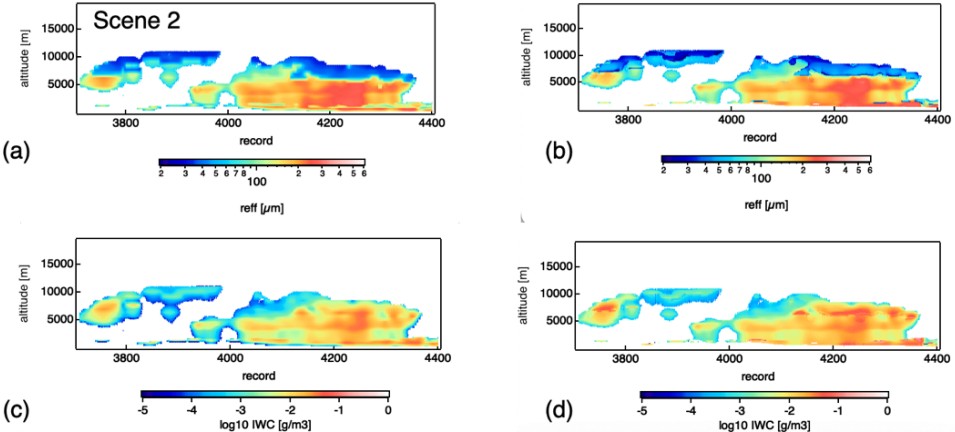

**Figure 5: Same as Fig.5 but for the ice microphysics corresponding to scene 2 in Fig.4 (b).**

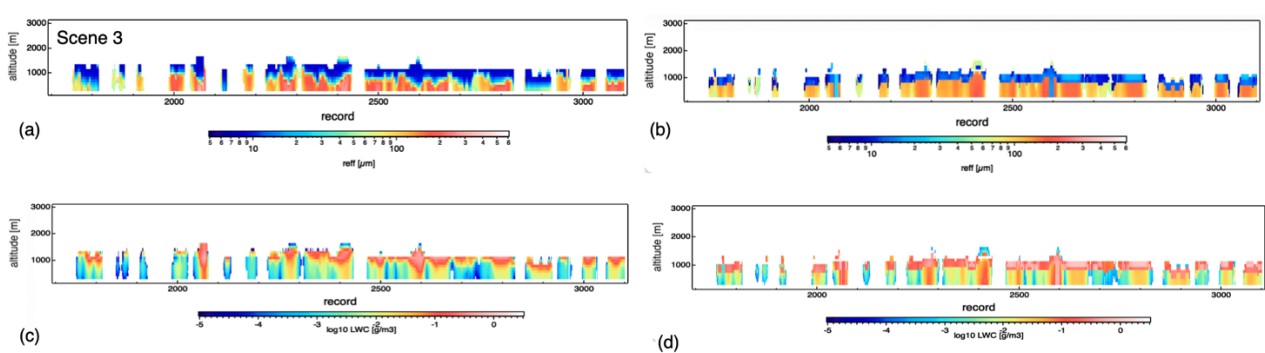

**Figure 6: Time–height cross-section of (a, c) simulated and (b, d) retrieved effective radius and total water content for a liquid-phase case corresponding to scene 3 in Fig.4 (c).**

We also performed a one-to-one comparison of retrieved (ret) and modeled (JS) effective radius ($r_{eff,ret}$ and $r_{eff,JS}$), ice water content ($IWC_{ret}$ and $IWC_{JS}$), and liquid water content ($WC_{ret}$ and $WC_{JS}$) at each JSG grid for AC_CLP using all 15 EarthCARE frames of the simulated observation data (Fig. 7, 8). The results showed that for both the ice and liquid phases, the majority of the retrieved population of $r_{eff}$ and water content lay close to the 1:1 line. For the ice phase, the slopes of the regression lines were generally around 0.8 and ice water content had about 14.5% mean relative error and tended to be

slightly overestimated when the ice water content were small (Fig. 7a). The effective radius of ice phase was evaluated at

small (Fig.7c) and large size ranges (Fig.7b) bounded at 60µm. In the model, three modes (i.e., ice cloud particles, snow, and graupel) contributes to the effective radius for the ice phase. Despite such complexity, the mean relative error of the effective radius retrievals for the larger size range was about 28.9%, and mean relative errors and the mean bias of the effective radius retrievals for the smaller size range were about 18.6% and 7.5%, respectively.

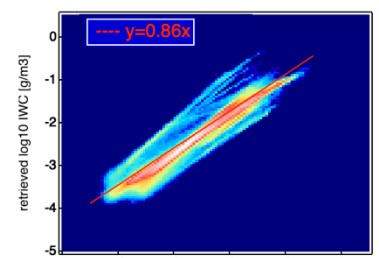 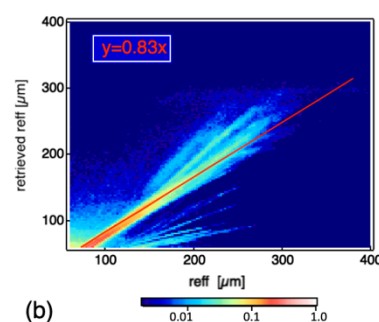 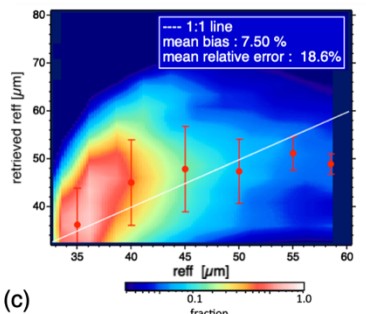

**Figure 7: Scatterplot of retrieved and modelled (a) ice water content and (b)(c) ice phase effective radius. Red and white solid lines correspond to a linear regression line forced through the origin and a 1:1 ratio, respectively. Symbols in (c) indicate means and standard deviation of retrieved $r_{eff}$.**

The liquid water content retrievals for the water clouds were able to track the change in the liquid water content and corresponded relatively well with the model truth. A larger scatter around the truth was observed at larger liquid water content range, which was biased low. Further analysis of the model suggested that this could in part occurred when liquid cloud particles made a major contribution to the water content, but a negligible contribution to $Z_e$ and its vertical structure,

which in some cases the algorithm made a slight misinterpretation when determining their contributions at the CPR-only regions. The slopes of the regression lines were also around 0.8 for both liquid water content and effective radius of liquid precipitation (Fig.8a, b). For the smaller size range, the frequency distribution of the retrieved effective radius was examined since the liquid cloud particles in the model has a constant effective radius of 8 µm (Fig.8c). It was seen that the retrieved peak size was close to the model truth and was around 10 µm, which was about 2 µm overestimation, and smaller fraction of

drizzle -sized particle was also retrieved. In future, we will further investigate the improvement in the microphysics retrieval when the ACM_CLP algorithm with CPR–ATLID–MSI synergy was applied to the simulated EarthCARE L1 data.

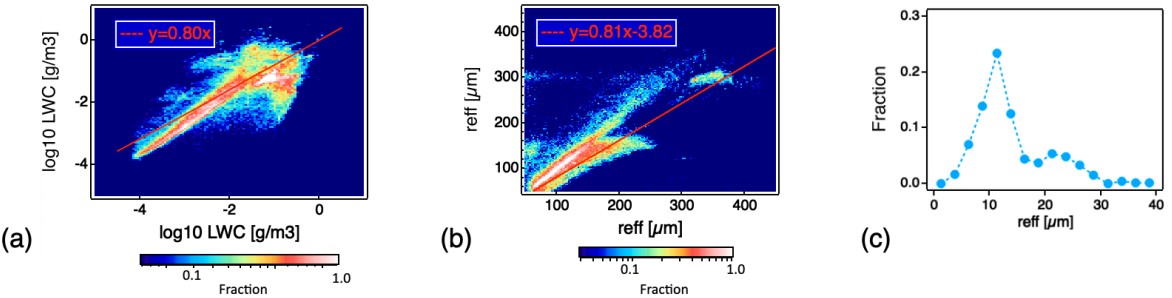

**Figure 8: Scatterplot of retrieved and modelled (a) liquid water content and (b) effective radius of liquid precipitation. Frequency distribution of the retrieved effective radius of cloud particles is shown in (c). Solid lines correspond to a linear regression line.**

## 4. Summary and Expectations

This study introduces the active sensor-based JAXA L2 cloud product, which is produced using three different algorithm-processing chains. The L2a CPR_CLP chain produces the standalone CPR cloud product; L2b AC_CLP produces the CPR–ATLID synergy cloud product; and L2b ACM_CLP produces the CPR–ATLID–MSI synergy cloud product. The cloud microphysics scheme considers the maximum of two different size distributions at each JSG grid to treat and capture the co-existence of cloud and precipitation particles or particles with different cloud phases. For the EarthCARE mission, the outputs from the JAXA L2 standard cloud product feature a 3D global view of the dominant ice habit categories and microphysics, and habit and size distribution transitions from cloud to precipitation. Demonstration of the JAXA L2 cloud particle category product using actual satellite data could show different preference for the occurrence of different ice habit categories. Cloud particle formation and growth conditions can be examined further by incorporating EarthCARE radar Doppler velocity measurements.

The active sensor-based JAXA L2 cloud products were assessed using simulated EarthCARE L1 orbit data created by the JAXA joint simulator, covering representative cloud and precipitation scenes over the globe. A comparison of retrieved and modelled microphysics obtained using the AC_CLP standard outputs as a reference showed that the retrieval reasonably reproduced the vertical profile of the modeled microphysics and the majority of the retrieved population of particle size and water content lay close to the 1:1 line with the slopes of the regression lines to be around 0.8 for both the ice and liquid phases. Velocity-related products from the JAXA L2 research cloud product and further improvements in the microphysics retrieval from the CPR–ATLID–MSI synergy will be reported in a future study.

In addition to assessing the L2 cloud product using simulated EarthCARE L1 data, ongoing studies will characterize the product in the framework of JAXA EarthCARE CAL/VAL activity. These studies include the use of ground-based radar and synergistic sensors at the NICT intensive observation site (Okamoto et al., 2024), complementary data from other spaceborne sensors such as A-TRAIN (Sato and Okamoto, 2023), and a European Union–Japanese collaboration to evaluate

CPR Doppler measurements and precipitation in CPR blind zones over Antarctica. As part of this joint activity, a validation methodology for space-borne Doppler radar was developed to obtain an unattenuated 94-GHz Doppler spectrum and related information on particle shape, sedimentation velocity, and size distribution at high temporal resolution from disdrometer and 24-GHz (K-band) Doppler radar synergy through frequency conversion and appropriate sampling strategies (Bracci et al., 2023). These validation datasets are highly valuable and will be used for further evaluation of the algorithms for EarthCARE,

launched on 28 May.

**Data availability.**

The JAXA L2 Echo products, ATLID products, and MSI products were processed by the National Institute of Information and Communications Technology (PI: Horie H.), National Institute for Environmental Studies (PI: Nishizawa, T.) and Tokai University (PI: Nakajima, Y. T.), respectively. The JAXA EarthCARE synthetic data are distributed from The standard products of the JAXA EarthCARE synthetic data are available from https://doi.org/10.5281/zenodo.7835229 (Roh et al., 2023). The CALIPSO Lidar Level 1B profile data V4-10 and CloudSat 2B-GEOPROF P1_R05 data used in this study are provided from the NASA Langley Research Center Atmospheric Science Data Center (https://doi.org/10.5067/CALIOP/CALIPSO/LID_L1-STANDARD-V4-10, NASA/LARC/SD/ASDC, 2016) and CloudSat Data Processing Center (https://www.cloudsat.cira.colostate.edu/data-products/2b-geoprof, Marchand & Mace, 2018), respectively. The KU CloudSat-CALIPSO Merged Data set is provided and updated to the latest version by JAXA A-Train Product Monitor (http://www.eorc.jaxa.jp/EARTHCARE/research_product/ecare_monitor_e.html).

**Author contributions.**

KS conducted research and drafted the paper. KS and HO developed the L2 cloud algorithms. TN, YJ, RK, TYN and MW processed the L1 data and produced the L2 products. MS and RW developed the EarthCARE synthetic data and provided the model outputs. HI provided scattering simulations. All provided useful discussions for the assessment of L2 cloud products.

**Competing interests.**

The authors have no competing interests to declare.

**Special issue statement.**

This article is part of the special issue "EarthCARE Level 2 algorithms and data products." It is not associated with a conference.

**Acknowledgements.**

The authors would like to thank the JAXA EarthCARE Science Team and the Remote Sensing Technology Center of Japan (RESTEC).

**Financial support.**

This study was supported by The Japan Aerospace Exploration Agency (EORA3) for the EarthCARE mission (grant no. 24RT000193); JSPS (KAKENHI Grants JP22K03721; JP24H00275); Research Institute for Applied Mechanics, Kyushu University (Fukuoka, Japan).

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
