# Peer review of "JAXA Level 2 cloud and precipitation microphysics retrievals based on EarthCARE radar, lidar and imager: The CPR\_CLP, AC\_CLP, ACM\_CLP products"

_Atmospheric Measurement Techniques, 2024_

## Referee Comment (RC2)

The paper provides a high-level summary of JAXA level 2 cloud and precipitation microphysical property products, which can help users effectively select suitable products for research and application in the future. The paper is well organized and presented. However, as I commented below, a few aspects could be improved.

Major issues:
1. EarthCARE radar provides Doppler velocity measurements, the sum of hydrometer falling speed and air vertical velocity. The potential of providing air vertical velocity estimation in convective clouds is exciting. The paper used several names to discuss air vertical velocity. For example, in the first paragraph, 'vertical velocity' and 'air motion' refer to the same parameter (to my understanding). But we think about 'air motion' in 3-D. In Table 1, you list the "Cloud air velocity" product, better called "Air vertical velocity." It will be great to use a consistent statement for retrieved air vertical velocity in the paper.

2. It would be beneficial to provide a paragraph or two in section 2.1 to place JAXA level 2 cloud products in the context of space-based multi-sensor cloud remote sensing and the reasoning for three cloud products. Although it is not possible to go into details of each algorithm, it could be helpful to provide a high-level summary of available information and challenges, general approaches, and additional information used to constrain retrievals to help readers better understand uncertainties in the products.

3. About processing flow (Section 2.2): The processing flow given in Fig. 1 is helpful in understanding the relationships among the three products. However, parameters under the two horizontal arrows could be better described in the text and positioned in the figure. In the summary, three processing chains (L2a, L2b, L2c) are mentioned but could be discussed in this section.

Minor issues:
1. Line 24: add " and cloud dynamics" after "hydrometer formation"
2. Line 42: Does "the EarthCARE L2" mean JAXA L2 here?
3. Line 102-104: This sentence could be incorrectly stated. Do you mean that ATLID-based results are used to train a CPR-based algorithm to provide retrievals in regions with CPR only measurements?
4. Line 129: "Eight frames" and "15 frames" are inconsistent here. One of the "frames" needs to be replaced with a different word.
5. In Figure 3, there are fewer clouds horizontally in simulated ATLID measurements, which is puzzling because ATLID should be more sensitive to CPR in cloud detection.
6. Figure 3 caption: add "(left column)" after "Ze measurements" and "(right column)" afte "product" to better separate CPR and ATLID measurements.
7. The layout of different panels between simulations and retrieval for Fig. 6 differs from Figs. 4 and 5. It would be better if they were consistent.
8. Line 182: Fig. 7b and Fig. 7c should be switched.
9. Line 218: change "Doppler information" to "radar Doppler velocity measurements".

---

## Author Comment (AC1)

The paper provides a high-level summary of JAXA level 2 cloud and precipitation microphysical property products, which can help users effectively select suitable products for research and application in the future. The paper is well organized and presented. However, as I commented below, a few aspects could be improved.

We are sincerely grateful to all reviewers for their careful reading of the manuscript and useful comments that have allowed us to improve its quality. In our revised manuscript, we have made corrections and added subsections in section 2 to provide a better overview of the products and major details of the algorithms and products based on comments by the reviewers. Below are the reviewers' comments in blue text followed by our replies in black text.

Major issues:
1. EarthCARE radar provides Doppler velocity measurements, the sum of hydrometer falling speed and air vertical velocity. The potential of providing air vertical velocity estimation in convective clouds is exciting. The paper used several names to discuss air vertical velocity. For example, in the first paragraph, 'vertical velocity' and 'air motion' refer to the same parameter (to my understanding). But we think about 'air motion' in 3-D. In Table 1, you list the "Cloud air velocity" product, better called "Air vertical velocity." It will be great to use a consistent statement for retrieved air vertical velocity in the paper.

We have improved the manuscript by using the term "air vertical velocity" throughout the paper and in Table 1.

2. It would be beneficial to provide a paragraph or two in section 2.1 to place JAXA level 2 cloud products in the context of space-based multi-sensor cloud remote sensing and the reasoning for three cloud products. Although it is not possible to go into details of each algorithm, it could be helpful to provide a high-level summary of available information and challenges, general approaches, and additional information used to constrain retrievals to help readers better understand uncertainties in the products.

Following the reviewer's suggestion, we have explained the rationale for using three cloud products and a high-level summary of the JAXA L2 cloud microphysics algorithms in Section 2.1. These additions to subsections 2.1.2 (Rationale for producing three products) and 2.1.3 (Summary of available information, challenges, general approaches, and additional information used to constrain retrievals) provide a better overview of the products. We have further improved the description of our general approach to microphysics retrieval in Section 2.3.

The following subsections has been added in the paper:

[revised manuscript text omitted]

3. About processing flow (Section 2.2): The processing flow given in Fig. 1 is helpful in understanding the relationships among the three products. However, parameters under the two horizontal arrows could be better described in the text and positioned in the figure. In the summary, three processing chains (L2a, L2b, L2c) are mentioned but could be discussed in this section.

Figure 1 has been improved, and corresponding text explaining the connections (inputs and outputs) of the three processing chains has been added to Section 2.2 (Processing flow of the JAXA Level 2 cloud microphysics product) as:

"The L2 cloud algorithms are processed in the following order: CPR_CLP, AC_CLP, and ACM_CLP. The cloud mask, cloud type, and cloud particle category products from each algorithm are passed to the high-order synergy algorithms. The CPR-only cloud mask, cloud type, and cloud particle category products from L2a CPR_CLP are input to the L2b AC_CLP algorithm, and these CPR-only derived products are combined with the ATLID-only cloud mask, cloud type, and cloud particle category to produce synergy CPR-ATLID products. These products are then applied to the AC_CLP algorithm to derive cloud microphysics products. The AC_CLP cloud mask, cloud type, and cloud particle category products are further passed to the ACM_CLP algorithm and used for 3-sensor microphysics retrieval. The MSI is not currently used to improve the cloud mask, type, and category products; therefore, these products from ACM_CLP are the same as those from AC_CLP."

Minor issues:

1. Line 24: add " and cloud dynamics" after "hydrometer formation"

   We added " and cloud dynamics".

2. Line 42: Does "the EarthCARE L2" mean JAXA L2 here?

   Yes. We corrected it to "EarthCARE JAXA L2".

3. Line 102-104: This sentence could be incorrectly stated. Do you mean that ATLID-based results are used to train a CPR-based algorithm to provide retrievals in regions with CPR only measurements?

   Yes. We have rephrased it as; ATLID-only CPC is used to train the CPR-based algorithm for ice particle category retrieval from $Z_e$ and temperature information in regions with CPR-only measurements. (Line 193-194)

4. Line 129: "Eight frames" and "15 frames" are inconsistent here. One of the "frames" needs to be replaced with a different word.

   Eight frames represent one orbit, and we used 15 frames for evaluation, corresponding to nearly two orbits. We have clarified this information in the text as, "The simulated L1 data for an EarthCARE orbit are divided into eight frames, and 15 frames, corresponding to nearly 2 orbits, are simulated to include representative cloud and aerosol scenes around the world." (Line 320-321)

5. In Figure 3, there are fewer clouds horizontally in simulated ATLID measurements, which is puzzling because ATLID should be more sensitive to CPR in cloud detection.

   The ATLID L2a cloud backscatter product for the cloud scenes in this study (Figure 3) is processed by the JAXA L2 ATLID algorithm (Nishizawa et al., 2024). Nishizawa et al., (2024) applied the JAXA ATLID L2 feature mask algorithm to the simulated EarthCARE L1 data and found that the cloud mask scheme appeared to reasonably extract cloudy pixels from the original output of ATLID signals produced by the model. The misidentification of the cloud layers was relatively low (approximately 10%). The effective radius/ice water content of the simulated ice clouds in Figure 3 were sometimes relatively large/small near cloud tops (Figures 4 and 5), and the corresponding ATLID backscattering coefficient could be weak to be detected.

Nishizawa, T., Kudo, R., Oikawa, E., Higurashi, A., Jin, Y., Sugimoto, N., Sato, K., and Okamoto, H.: Algorithm to retrieve aerosol optical properties using lidar measurements on board the EarthCARE satellite, Atmos. Meas. Tech. Discuss. [preprint], https://doi.org/10.5194/amt-2024-100, in review, 2024.

6.  Figure 3 caption: add "(left column)" after "Ze measurements" and "(right column)" afte

    "product" to better separate CPR and ATLID measurements.

    We included them.

7.  The layout of different panels between simulations and retrieval for Fig. 6 differs from Figs.

    4 and 5. It would be better if they were consistent.

    The layout of Fig.6 is modified to be consistent with Figs.4 and 5.

8.  Line 182: Fig. 7b and Fig. 7c should be switched.

    Fig. 7b and Fig. 7c are switched.

9.  Line 218: change "Doppler information" to "radar Doppler velocity measurements".

    We have changed it.

    Thank you for your suggestions.

---

## Author Comment (AC2)

**RC1**: 'Comment on amt-2024-99', Anonymous Referee #1, 03 Jul 2024

This paper provides a brief overview of the radar-only, radar-lidar and radar-lidar-radiometer cloud and precipitation retrieval products from the JAXA Level 2 production model, with illustrations of key variables based on numerical forecast models and CloudSat/CALIPSO. The results in this paper are well-presented and summarises a long record of work with CloudSat/CALIPSO and other instruments in preparation for EarthCARE. While we appreciate concise papers, ultimately the paper does not provide enough of a detailed description of the algorithm(s), nor a consistent view of the data products in question. I recommend this paper for major revisions to address these issues.

We are sincerely grateful to all reviewers for their careful reading of the manuscript and useful comments that have allowed us to improve its quality. In our revised manuscript, we have made corrections and added a better overview and major details of the algorithms and products based on comments by the reviewers. Below are the reviewers' comments in blue text followed by our replies in black text.

Major comments

1. The names of the products (i.e. C-CLP, AC-CLP, ACM-CLP) have typically been given in the paper titles within this special issue, and this would help the user to navigate through the special issue as well as the EarthCARE L2 data products.

   Product names have been added to the title as follows; JAXA Level 2 cloud and precipitation microphysics retrievals based on EarthCARE CPR, ATLID, and MSI: The C-CLP, AC-CLP, ACM-CLP products

2. There's a fundamental ambiguity in the Abstract, which isn't resolved within the paper. We read that, (L16-17) "these products provide a detailed view of [cloud properties] as well as vertical velocity information", but then in the final sentence, (L20) "Level 2 velocity-related products will be described in a future paper." Please make it very clear from the abstract onward that the vertical velocity products are not described in the present paper.

   We have deleted "as well as vertical velocity information" (L16–17) and edited the final sentence to read: JAXA Level 2 velocity-related products (i.e., CPR_VVL, AC_VVL, and ACM_VVL) such as hydrometeor fall speed and air vertical velocity will be described in a future paper. (L21-22)

3. Table 1 is key to detailing the production model represented within this paper. The "standard" version of each L2 product are given, and also "research" versions of each product which contain vertical velocity information (i.e. retrieved precipitation mass flux, in-cloud vertical air velocity). This is ambiguous: are the existing products to be updated at a later date to include these additional variables, or are additional products to be released? Asked another way: it's not clear from Table 1 or from the text of this paper whether the JAXA L2 production model as described in Eisinger et al. (2024) is accurate: in Figure 3 of Eisinger et al (2024) the C-CLP, AC-CLP and ACM-CLP products will provide the "standard" quantities relating to cloud mask, phase, shape and cloud microphysics, while additional "research" L2 data products with different names will provide complementary

information: e.g. C-RAS for CPR retrievals of rain and snow properties, C-VVL for vertical velocity; ACM-ICE for ice cloud effective radius as described in Eisinger et al. (2024). In Table 1, the titles of the three last columns indicate that all of these variables fall under "C-CLP", "AC-CLP" and "ACM-CLP", and no other products are named. Please resolve this ambiguity, while citing the Eisinger et al. (2024) paper which was intended as a centralised resource for understanding the full range of products available.

The JAXA L2 products summarized by Eisinger et al. (2024) are correct. Table 1 and the corresponding text have been updated for consistency with Eisinger et al. (2024). The main research products (C-RAS, AC_RAS, ACM_RAS, CPR_VVL, AC_VVL, and ACM_VVL products) have been added to Table 1 and the text. The C-CLP, AC-CLP, and ACM-CLP products include both cloud and precipitation microphysics, but are reported only as cloud microphysics, whereas C-RAS, AC_RAS, and ACM_RAS include precipitation-only products using Doppler velocity. The C-CLP, AC-CLP, and ACM-CLP products will eventually be updated using Doppler velocity. For ACM_CLP, this update will be included in ACM_CDP, which is processed by JAXA Laboratories (LR), but results fulfilling the release criteria could be added to the ACM-CLP products. ACM-ICE is under consideration.

Subsection 2.1.1 is added in the paper as follows;

"2.1.1 Primary cloud products

Standard cloud property (CLP) products (i.e., CPR_CLP, AC_CLP, and ACM_CLP) include a cloud mask, cloud particle type, cloud particle habit category, cloud microphysics, cloud optical thickness, and cloud water/ice paths (Table 1). The microphysical properties of all hydrometeor types in the standard products are reported in the cloud microphysics product, and precipitation-sized particles are not separated into precipitation products. JAXA L2 research cloud products include velocity-related products such as sedimentation velocity and air vertical velocity (Sato et al., 2009), which are designated CPR_VVL, AC_VVL, and ACM_VVL; precipitation-only products (e.g., rain and snow rates; CPR_RAS, AC_RAS, and ACM_RAS) (Table 1). Details of these research products will be reported in a future paper. All products are reported using the Joint Standard Grid (JSG) with 1-km horizontal and 100-m vertical grid spacing. Note that CPR_CLP, AC_CLP, and ACM_CLP are produced with and without the use of L2 CPR Doppler velocity to show the effect of additional information obtained from Doppler velocity. The version without Doppler velocity will eventually be updated based on the version using Doppler velocity. Similarly, research products will be developed through RAS and VVL, and results fulfilling the release criteria may be added to the standard products (i.e., CPR_CLP, AC_CLP, and ACM_CLP) for release."

4. The algorithm description section 2.3 of this paper is very dense and difficult to understand.

- It begins by citing the ATBD, but a link to this document is not provided—and in any case, I would ask that the authors provide some recapitulation of the main detail provided in the ATBD within this paper, or in the

citations within. As in other papers in this special issue, this should provide at least an overview of the algorithm, the key physical assumptions, etc.

The descriptions of the L2 cloud algorithms in section 2.3 have been improved to provide a better overview and major details of the algorithms. The reference to ATBD has been replaced by Okamoto et al. (2024b), which is a better reference for the interrelations among algorithms and products, corresponding to the subjects covered in this paper. A general summary of the JAXA L2 cloud microphysics algorithms is added in subsection 2.1.3

• It would be helpful to break the description into paragraphs at least, but even more helpfully into sub-sections: cloud mask, phase discrimination, cloud microphysics. Here it would also be useful to provide a description of the intended use of Doppler measurements for vertical velocity products, even if they are to be more fully described in a later paper.

Thank you for your suggestion.

The text in section 2.3 has been divided into subsections related to the preprocessing algorithms for cloud microphysics retrieval (i.e., subsection 2.3.1 Preprocessing for cloud microphysics retrieval; 2.3.1.1 Cloud mask; 2.3.1.2 Cloud type; 2.3.1.3 Cloud particle category (CPC)), and cloud microphysics retrieval algorithms (i.e., 2.3.2 Cloud microphysics; 2.3.2.1 Ice cloud microphysics; 2.3.2.2 Liquid cloud microphysics).

The intended use of Doppler velocity has been added to subsections 2.3.1 and 2.3.2 for cloud preprocessing and microphysics retrieval and subsection 2.3.3 for air vertical velocity (i.e., 2.3.3 
[revised manuscript text omitted]

Minor comments

• L91: please provide a DOI for the ATBD

The reference to ATBD has been replaced by Okamoto et al. (2024b) (reply to major comment 4). This reference is cited in section 2.2.

• L119: what are the two different size distributions?

The description for particle size distributions is provided in subsection 2.3.1 in the revised manuscript. A modified gamma size distribution is assumed for cloud ice and snow, and a log-normal size distribution is assumed for warm water, super-cooled liquid, and warm precipitation. For both ice- and liquid-phase clouds, a maximum of two different particle size distributions can be considered within one JSG grid to handle the presence of multiple cloud modes (i.e., cloud ice, cloud water, or super-cooled water), precipitation modes (drizzle, rain, or snow), and

cloud particles of differing phases. Therefore, two different effective radii with corresponding ice water or liquid water content and other microphysical properties are derived for each active sensor grid within a vertical profile. The retrieval procedure of the two size modes are provided in subsections 2.3.2.1 and 2.3.2.2.

---

## Author Response (AR2)

**Manuscript amt-2024-99**
**Response to Reviewers**

Dear Editor Dr. Hogan,

Thank you for giving us the opportunity to submit a revised version of the manuscript to Atmospheric Measurement Techniques. We are grateful to all reviewers for their constructive comments and suggestions that have allowed us to improve the quality of the manuscript. In our revised manuscript, we have made corrections based on comments by Reviewer 2 and the title is improved as suggested by the Editor.

Here is our point-by-point response to the reviewers' comments.

**Author's response to reviewer report 1**

This manuscript has been improved and clarified by the addition of further details about the context of this products within the JAXA production model, the future plans for the addition of Doppler velocity, and the detailed descriptions of both the preprocessing and the microphysical assumptions used within these retrievals. We thank the authors for their very thorough response to feedback.
I now recommend this paper for publication.

Thank you very much for your great efforts in reviewing our paper.

**Author's response to reviewer report 2**

The revised paper is significantly improved. But there are a few minor issues need to be addressed before publication.
1.  The abbreviations of products (C-CLP, AC-CLP, and ACM-CLP) could be defined better in the abstract and text. But, the C-CLP, used in the title and abstract, was not used in the text (was replaced by CPR_CLP). Also, AC-CLP, and ACM-CLP in the title and abstract were written as AC_CLP, and ACM_CLP.

**Response:** Thank you for pointing this out. C-CLP, AC-CLP and ACM-CLP have been corrected to CPR_CLP, AC_CLP, ACM_CLP, respectively. We have included the definitions of C-CLP, AC-CLP and ACM-CLP in the abstract (Lines 16-17) and text (Lines 55-56) as follows;

Abstract (Lines 16-17)
"i.e., The cloud radar standalone cloud product (CPR_CLP), the radar-lidar synergy cloud product (AC_CLP), and the radar-lidar-imager cloud product (ACM_CLP))."

Section 2.1.1 (Lines 55-56)
"Standard cloud property (CLP) products (i.e., CPR standalone CPR_CLP product, CPR-ATLID synergy AC_CLP product, and CPR-ATLID-MSI synergy ACM_CLP product) include …."

2. Line 16-17: here only two instruments are mentioned here. For ACM-CLP, you need MSI, which is not defined in the paper.

    **Response:** MSI is added to Lines 17-19 as,

    "Combined with 94-GHz Doppler cloud profiling radar (CPR), 355-nm high-spectral-resolution lidar (Atmospheric Lidar: ATLID) and Multi-Spectral Imager (MSI),…"

3. Lines 103-104: Cloud particles and large particles referred here are confusion. Cloud particles have different sizes, thus, there are small and large even in a single-phase clouds. When you consider ice and liquid phase for a gird, the ice phase has large size than liquid phase. Furthermore, precipitation particles are larger than cloud particles.

    **Response:** Thank you for your comment. The text (Lines 104-106) has been corrected to,

    "$Z_e$ is less sensitive to cloud particles in the presence of precipitation particles in ice- or liquid clouds, and $Z_e$ is less sensitive to liquid cloud particles in the presence of ice particles in mixed phased clouds. In such cases, …."

4. Line 125: Should '"inputs" be "outputs".

    **Response:** We have corrected it (Line 128). Thank you.

5. Figure 1: The symbols given in Fig. 1 need to be defined. in the text or figure caption.

    **Response:**

    We have provided the definition of the symbols in the figure caption as follows;

    "JAXA L2 Echo product contains the radar reflectivity factor ($Z_e$), Doppler velocity ($V_D$), normalized radar cross-section ($\sigma_0$), pulse integrated attenuation (PIA). JAXA L2 ATLID product contains the extinction coefficient ($\alpha_{ext}$), attenuated ($\beta_{att}$) and true backscattering coefficient ($\beta$), and depolarization ratio $\delta$."

    In addition to the above comment, we have corrected "Multi-Spectral Information" in the figure caption to "Multi-Spectral Imager (Line 155) and in Table 1. (Line 124)".

    Thank you for your useful suggestions.